# Swimmer’s Shoulder: Ultrasound Anatomical Description of Shoulder Tendons in Elite Swimmers and Water Polo Players

**DOI:** 10.3390/healthcare13020151

**Published:** 2025-01-14

**Authors:** Beatriz Arrillaga, Albert Pérez-Bellmunt, Iker García, Luis Rodríguez-Adalia, Ingrid Möller, Maribel Miguel-Pérez

**Affiliations:** 1Unit of Human Anatomy and Embryology, Department of Pathology and Experimental Therapeutics, Faculty of Medicine and Health Sciences, University of Barcelona, L’Hospitalet de Llobregat, 08907 Barcelona, Spain; beatrizarrillaga@hotmail.com (B.A.); ingridmoller@ub.edu (I.M.); 2ACTIUM Research Group, Department of Basic Sciences, International University of Catalonia, 08017 Barcelona, Spain; aperez@uic.cat; 3Department of Cellular Biology, Physiology and Immunology, Physiology Section, Faculty of Biology, University of Barcelona, 08028 Barcelona, Spain; ikergarciaalday@gmail.com; 4Grup de Recerca en Ciències de l’Esport INEFC Barcelona (GRCEIB), National Institut of Physical Education of Catalonia, 08038 Barcelona, Spain; rodriguezadalia@gmail.com; 5Poal Institute of Rheumatology and Physiotherapy, 08022 Barcelona, Spain

**Keywords:** rotator cuff, shoulder pain, supraspinatus, swimming, ultrasonographic diagnosis

## Abstract

**Objectives:** This study aimed to examine the differences in shoulder ultrasonographic parameters between overhead aquatic athletes and their relationship with shoulder pain. **Methods:** Sixty-four healthy junior subjects (*n* = 128 shoulders) participated in the study, including 17 healthy controls, 25 elite junior swimmers, and 22 elite junior water polo players. An ultrasonographic study of the shoulder was carried out to measure the thickness of the rotator cuff structures in sex- and age-paired groups. **Results:** Compared to controls, female and male swimmers and water polo players had thicker values in the rotator cuff tendons, including the subscapularis (SSB) (*p* < 0.001), supraspinatus (*p* < 0.001), infraspinatus (*p* < 0.001) and teres minor (Tm) (*p* < 0.025). The female swimmers had thicker values than the water polo players in the SSB (*p* < 0.001) and Tm (*p* < 0.011). In the comparison by sexes, the female controls had a thinner LHBB, SSB and Tm (*p* < 0.019), the female swimmers had a thinner ISP (*p* = 0.001), and the female water polo players had a thinner LHBB, SSB, ISP and acromiohumeral distance than their group-paired counterparts (*p* < 0.018). Finally, the females had a lower incidence of positive Jobe test results compared to the males (*p* = 0.018), which was associated with lower pain scores at rest (*p* < 0.034) and during training (*p* < 0.036). **Conclusions:** The rotator cuff tendon structures were larger in the healthy overhead aquatic athletes compared to the age- and sex-paired controls. The females had smaller values in the shoulder ultrasonographic tendon parameters compared to the age- and sport-paired males, except for the supraspinatus tendon.

## 1. Introduction

Swimming and water polo are popular recreational and competitive aquatic disciplines that include the swimming stroke during their practice [1]. The shoulder is the most demanded joint in both sports [2,3], making shoulder pain the most prevalent injury in these overhead disciplines [4,5]. Repetitive movement of the swimming stroke, including overhead flexion, abduction, and internal rotation, is related to shoulder pain, representing typical impingement positions [6,7]. In water polo, the shoulder also suffers from microtrauma due to the combination of swimming and overhead throwing [2]. This sporting gesture increases the range of motion and may induce laxity in the passive structures of the shoulder, leading to an increased risk of soft tissue injury and pain [3,4,7].

Swimmers have a 40–91% incidence rate of shoulder pain [8,9], while water polo players report up to 80% [10]. Shoulder pain is so common in these sports that, as early as 1974, the term “swimmer’s shoulder” was first described as a “common painful syndrome of repetitive shoulder impingement in swimmers” [5]. Some authors include subacromial impingement syndrome (SIS), supraspinatus (SSP) tendinopathy, subacromial bursitis, rotator cuff (RC) tendinosis, long-head biceps brachii (LHBB) tendinosis, and glenohumeral multidirectional hypermobility [6,11,12,13].

Researchers identified two intrinsic mechanisms for this condition. The primary mechanism proposed in 1972 is tendon impingement under the coracoacromial arch [14]. However, this is less common in swimmers than the secondary mechanism [15], characterized by sports stress-induced glenohumeral instability and fatigue in the static and dynamic stabilizers, which lead to RC tendinopathy [14]. Fatigue affects swimming technique in swimmers with shoulder pain, compared to those without, resulting in friction of the SSP tendon [15]. The modified activation pattern observed in swimmers experiencing shoulder pain has been associated with an augmentation in the thickness of the SSP tendon subsequent to a training session [8]. Following these results, after a fatigue protocol, the increase in SSP tendon thickness persists for up to 24 h later [16]. Similarly, this instability and the consequent impingement can be related to the dysfunction of the scapulothoracic muscles or a muscle imbalance caused by the sporting gesture itself [14]. When considering this in conjunction with the observed correlation between the years of competitive participation and the augmentation in the thickness of tendons, including the SSP, it is important to emphasize the need for regular, individualized shoulder monitoring for each athlete. This is crucial for clinical evaluators and sports technical staff [11,17].

Imaging techniques such as ultrasound (US) are a reliable method for diagnosing shoulder injuries and investigating tendon characteristics, such as the measurement of SSP tendon thickness and acromiohumeral distance (AHD), which can inform on the presence of impingement syndrome or RC tendinopathy [8,14]. This measurement method is useful to determine the potential for injury and to monitor training load, as some studies have reported a relationship between AHD and SSP tendon injury [1,17,18]. There is no consensus on the normal thickness of the shoulder tendons in overhead athletes. Some studies have described that the SSP tendon without pathology has a thickness of 6 mm [17,18,19], while other studies have established a difference of 0.6–1.5 mm in the thickness of the SSP tendon between shoulders with and without pain [17]. In line with these findings, other authors have also observed that individuals with subacromial impingement syndrome have a thicker SSP tendon [17] and the existence of an increased tendon thickness of the subscapularis (SSB) tendon in swimmers [1]. However, studies that examine all shoulder RC tendons and the thickness of their fascia in sports such as swimming or water polo are lacking, as well as its relationship with shoulder pain.

Given the high rates of shoulder injuries in swimming and water polo [20], examination of the RC tendons and the effect of swimming practice may help to elucidate the role of RC in the etiology of shoulder pain. Similarly, measuring and standardizing its thickness can help to monitor training loads, thus acting to prevent injuries. Currently, there is no consensus on the standard shoulder tendon thickness for overhead athletes, nor established differences by gender or age, which could impact clinical practices for coaches and healthcare professionals. This study aimed to examine the ultrasonographic differences in the RC tendon thickness between swimmers, water polo players and control populations, and to assess their relationship with shoulder pain, in sex- and age-paired groups. The findings of this study will provide reference values that can serve as a foundation for clinical evaluations and follow-ups aimed at the prevention of shoulder injuries in athletes participating in overhead sports.

## 2. Materials and Methods

### 2.1. Participants

A sample size of 64 healthy junior subjects (*n* = 128 shoulders) was invited to participate in the study, including 17 healthy controls (F = 9; M = 8); 25 elite junior swimmers (F = 15; M = 10) and 22 elite junior water polo players (F = 13; M = 9). Measurements were carried out during the preseason period at the Sant Cugat High-Performance Center under the Catalan Swimming Federation training program. The entire sample consisted of 37 females (*n* = 74 shoulders) and 27 males (*n* = 54 shoulders), with an average age of 15.6 ± 1.0 years. Additional descriptive parameters of the sample are compared at the beginning of the Results section.

Swimmers performed 25 h of training per week in 11 sessions of swimming training, in which they performed a weekly volume of 35 to 45 km in this phase of the season (which can increase up to 65 km weekly in further periods of the season), and 5 dry-land training sessions. Swimmers had an average of 707 FINA points (that is, a value of the best swimmer’s mark relative to the world’s best mark). Water polo players performed 25 h of training per week in 11 training sessions per week, including the match, in which they performed 20 to 25 km of swimming training and additional sport-specific exercises. The aquatic athletes were considered at the elite junior level, with 18 out of 47 athletes representing the junior national team in international competitions, being 9 of them World or European junior medalists. Despite the absence of age distinction in the Participant Classification Frameworks, the participants can be considered “Tier 4: Elite/International level”, since they are likely to be considered at the National (Junior) Team, some of them has been International or Continental (Junior) medalists, and they are part of a training program with intention to complete at top-level competition [21].

In the case of aquatic athletes, inclusion criteria were (1) having more than 5 years of aquatic competitive training experience, (2) being part of the same training group in each aquatic sport, and (3) being within the junior category, in case of swimmers, or juvenile category, in case of water polo players, of the assigned sport. Exclusion criteria were (1) not participating in any overhead competitive aquatic sport and (2) not having had any previous shoulder surgery in their lifetime, or any previous shoulder injury in the last 6 months.

In the case of controls, inclusion criteria were (1) to have a similar age, sex, and BMI than aquatic athletes, and (2) to be apparently healthy. Exclusion criteria were (1) to have practiced any overhead competitive sport during the previous 5 years, and (2) to have had any previous shoulder surgery in their lifetime or any previous shoulder injury in the last 6 months.

All participants were informed about the purpose, protocol, and procedures before informed consent was obtained from them or their representatives. The principles of the Declaration of Helsinki for human experimentation were followed at all times, while ethical approval was obtained from the corresponding Institutional Review Board (IRB00003099).

### 2.2. Experimental Design and Procedures

Ultrasound was performed with an Alpinion E-Cube i7 (Alpinion Medical Systems, Gangseo-gu, Seoul, Republic of Korea), equipped with a linear electronic transducer, with a frequency range of 3 to 12 MHz. Bone reference was to carry out the measurements, and the same configuration was maintained in the ultrasound machine, only varying the focus or depth as needed. First, a pilot study was conducted with three experienced testers to establish the correlation index, and then the measurements were performed by the same examiner three times, following the location guidelines established for the study (Table 1). The average of the two closest measurements was used for further analysis. Through the US technique, the thickness of the LHBB tendon and the RC tendons, including subscapularis (SSB), SSP, infraspinatus (ISP) and teres minor (Tm) tendons, were measured.

Subsequently, the SSP, ISP, and Tm fascia thicknesses were measured and finally, the AHD occupation ratio (OR; OR = SSP/AHD) was calculated. Both shoulders were evaluated in each. Finally, all subjects completed the Visual Analog Scale (VAS) and underwent the Jobe test. 

### 2.3. Statistical Analysis

Data are reported as mean values ± standard deviation. The Shapiro-Wilk test was used to establish the normal distribution of the sample. Differences in the shoulder parameters of ultrasonography (as dependent variables) and pain scales were analyzed using a two-way repeated measures ANOVA with two independent factors: sex (females and males) and group (controls, swimmers and water polo players). In the case of detecting statistical effects (*p* < 0.05), Bonferroni comparisons were made. Effect sizes as partial eta squared (ηp2\eta^2^_pηp2) values were employed to present the magnitude of differences, with thresholds of 0.01, 0.06 and above 0.14 for small, medium and large effects, respectively [24]. Statistical power was also calculated.

Within-group differences between breathing and non-breathing sides in swimmers, between throwing and non-throwing arms in water polo players, and differences in pain scales according to the Jobe test result (positive or negative) were evaluated using Student’s t-test. Effect size (Cohen d) was calculated to estimate the magnitude of the difference between group means, with d = 0.1, 0.3, 0.5, 0.7, and 0.9 reflecting the effect sizes small, medium, large, very large, and extremely large effect sizes, respectively [25].

The association between shoulder pain scales and ultrasonographic parameters of aquatic athletes (swimmers and water polo players) was explored by Pearson’s linear regression analysis (measures the relationship between continuous variables). These were carried out separately for females and males. Finally, a chi-square independence test was performed (assesses the association between categorical variables) to examine the relationship between sex and Jobe test results in the sample of aquatic athletes. The software package used for the statistical analysis was SPSS v26 (IBM SPSS Statistics, Armonk, NY, USA).

## 3. Results

The reliability of the shoulder measurements was evaluated using the intraclass correlation coefficient (ICC). The inter-rater correlation was conducted in a preliminary sample of 11 water polo players (*n* = 22 shoulders). The results of the study’s lead technician for all the ultrasonographic shoulder parameters were compared with two senior technicians of the research group (ICC 3, 1, two-way mixed, absolute agreement) with results ranging from 0.636 to 0.934, indicating moderate to excellent reliability.

### 3.1. Description of the Studied Participants

The study participants were paired by age and described by sex. In females, there were no statistically significant differences in the age between controls, swimmers and water polo players respectively (15.9 ± 0.9 vs. 15.4 ± 1.0 vs. 15.1 ± 1.4 years; *p* > 0.062). Weight was not significantly different between female controls, swimmers and water polo players respectively (58.0 ± 8.6 vs. 59.2 ± 3.7 vs. 60.5 ± 4.5 kg; *p* > 0.636). Regarding height, female controls were smaller than female swimmers and water polo players respectively (161.9 ± 5.6 vs. 169.0 ± 3.9 vs. 170.1 ± 6.0 cm; *p* < 0.001).

In males, there were no statistically significant differences in the age between controls, swimmers and water polo players respectively (15.8 ± 0.7 vs. 15.8 ± 0.8 vs. 16.3 ± 0.7 years; *p* > 0.057). Weight was not significantly different between male controls, swimmers and water polo players respectively (73.0 ± 11.9 vs. 70.6 ± 9.7 vs. 79.9 ± 14.0 kg; *p* > 0.195). Regarding height, male controls were smaller than male swimmers (178.0 ± 7.2 vs. 182.1 ± 7.8; *p* = 0.004), while water polo players were not different from their counterparts (186.5 ± 6.3 cm; *p* > 0.244).

### 3.2. Sex Differences in the Ultrasonographic Shoulder Parameters in Controls, Swimmers and Water Polo Players

The differences between sexes in the ultrasonographic measurements of the tendons of the LHBB and the RC muscles are shown in Table 2.

There was a significant interaction between the LHBB tendon and the sex of the participants (F = 17.210, *p* = 0. 001, ŋ^2^p = 0.534, sp = 0.972). Female controls (2.72 ± 0.60 mm, *p* = 0.009) and water polo players (2.63 ± 0.61 mm, *p* = 0.001) had lower values in the LHBB tendon compared to male controls and water polo players (3.35 ± 0.45 mm and 3.63 ± 0.69 mm, respectively).

There was a significant interaction between the SSB tendon and the sex of the participants (F = 26.700, *p* < 0. 001, ŋ^2^p = 0.640, sp = 0.998). Female controls (3.43 ± 0.68 mm, *p* = 0.003) and water polo players (4.79 ± 0.83 mm, *p* = 0.001) had lower values in the SSB tendon compared to male controls and water polo players (4.27 ± 0.43 mm and 6.27 ± 0.87 mm, respectively).

There was a significant interaction between the ISP tendon and the sex of the participants (F = 33.396, *p* < 0. 001, ŋ^2^p = 0.690, sp = 1.000). Female swimmers (5.58 ± 0.77 mm, *p* = 0.001) and water polo players (5.48 ± 0.81 mm, *p* = 0.018) had lower values in the ISP tendon compared to male swimmers and water polo players (6.63 ± 1.04 mm and 6.21 ± 1.14 mm, respectively).

There was a significant interaction between the Tm tendon and the sex of the participants (F = 12.064, *p* = 0. 003, ŋ^2^p = 0.446, sp = 0.900). Female controls (2.58 ± 0.39 mm, *p* = 0.019) had lower Tm tendon compared to male controls (2.92 ± 0.57 mm).

There was a significant interaction between AHD and the sex of the participants (F = 6.196, *p* = 0.025, ŋ^2^p = 0.292, sp = 0.644), in which female water polo players had lower values than males (12.30 ± 1.92 mm and 13.48 ± 1.80 mm respectively, *p* = 0.007).

Analysis of the SSP tendon, the OR, and the SSP, ISP and Tm fascia thicknesses showed no statistically significant differences between females and males in any of the studied groups (*p* > 0.142).

### 3.3. Group Differences in the Ultrasonographic Shoulder Parameters of Females and Males

Differences in the ultrasonographic shoulder parameters between controls, swimmers and water polo players are presented in Figure 1 (females) and Figure 2 (males).

Regarding the SSB tendon, there was a significant interaction between the studied groups (F = 70.259, *p* < 0. 001, ŋ^2^p = 0.824, sp = 1.000). Female and male swimmers (5.81 ± 0.91 mm, 6.91 ± 1.36 mm, *p* < 0.001) and water polo players (4.79 ± 0.83 mm, 6.27 ± 0.87 mm, *p* < 0.001) had higher values in SSB tendon compared to controls (3.43 ± 0.68 mm, 4.27 ± 0.43 mm), while female swimmers had thicker values than female water polo players (5.81 ± 0.91 mm vs. 4.79 ± 0.83 mm, *p* < 0.001).

There was a significant interaction between SSP and the studied groups (F = 30.365, *p* < 0.001, ŋ^2^p = 0.669, sp = 1.000). Female and male swimmers (6.24 ± 1.06 mm, 6.81 ± 1.30 mm, *p* < 0.001) and water polo players (5.99 ± 0.84 mm, 6.31 ± 1.35 mm, *p* < 0.001) had higher values in SSP tendon compared to controls (4.47 ± 0.50 mm, 4.67 ± 0.95 mm).

There was a significant interaction between ISP and the studied groups (F = 42.757, *p* < 0.001, ŋ^2^p = 0.740, sp = 1.000). Female and male swimmers (5.58 ± 0.77 mm, 6.63 ± 1.04 mm, *p* < 0.001) and water polo players (5.48 ± 0.81 mm, 6.21 ± 1.14 mm, *p* < 0.001) had higher values in SSP tendon compared to controls (4.05 ± 0.66 mm, 4.46 ± 0.60 mm).

Regarding the Tm tendon, there was a significant interaction with the studied groups (F = 23.796, *p* < 0. 001, ŋ^2^p = 0.613, sp = 1.000). Female and male swimmers (0.50 ± 0.14 mm, 0.48 ± 0.12 mm, *p* < 0.002) and water polo players (0.53 ± 0.10 mm, 0.51 ± 0.15 mm, *p* < 0.025) had higher values in the Tm tendon compared to controls (0.55 ± 0.11 mm, 0.61 ± 0.14 mm), while female swimmers had thicker values than female water polo players (0.50 ± 0.14 mm vs. 0.53 ± 0.10 mm, *p* < 0.011).

There was a significant interaction between AHD and the studied groups (F = 16.570, *p* < 0.001, ŋ^2^p = 0.525, sp = 0.999). Male swimmers (13.39 ± 2.01 mm, *p* < 0.001), and female and male water polo players (12.30 ± 1.92 mm, 13.48 ± 1.80 mm) had higher values than controls (10.62 ± 2.31 mm, 10.58 ± 2.11 mm, *p* < 0.019).

Analysis of the LHBB tendon, the OR, and the SSP, ISP and Tm fascia thicknesses showed no statistically significant differences between swimmers, water polo players and controls (*p* > 0.052).

### 3.4. Breathing and Non-Breathing Side Comparison in the Swimmer’s Ultrasonographic Parameters

Table 3 shows the comparison in ultrasonographic measurements between the breathing and non-breathing sides in swimmers with no statistically significant differences in any ultrasonographic shoulder parameter.

### 3.5. Throwing and Non-Throwing Arm Comparison in the Water Polo Player’s Ultrasonographic Parameters

Table 4 shows the comparison in ultrasonographic measurements between throwing and non-throwing arms in water polo players. Males had higher values in the LHBB tendon thickness of the non-throwing arm (4.02 ± 0.50 mm, *p* = 0.012; d = 1.332), and females had higher values in the Tm tendon thickness of the non-throwing arm (3.53 ± 0.56 mm, *p* = 0.034; d = 0.950).

### 3.6. Association Between Pain Scales and Ultrasonographic Shoulder Parameters in Aquatic Athletes

The VAS and ultrasonographic shoulder parameters were analyzed considering the total sample of aquatic athletes, including swimmers and water polo players to ensure an adequate statistical analysis due to the high values of standard deviation in the pain scales. Additionally, the statistical analysis of the Jobe test was performed on the entire sample of aquatics athletes due to the low prevalence of a positive test in female swimmers (*n* = 3) and water polo players (*n* = 6).

### 3.7. VAS

In the comparison between sexes, VAS at rest was not different between female and male aquatic athletes (1.32 ± 2.02 vs. 1.03 ± 1.47; *p* = 0.456). Regarding VAS during training, there were no differences between female and male aquatic athletes (2.43 ± 2.53 vs. 3.31 ± 2.64; *p* = 0.140). The associations between VAS at rest, or VAS during training and the ultrasonographic shoulder parameters were not statistically significant in any measurement (*p* > 0.068).

### 3.8. Jobe Test

A chi-square test of independence was performed to examine the relationship between sex and the result of the Jobe test in aquatic athletes. Females were more likely to have a negative result on the Jobe test than males, X2 (1, N = 74) = 5.60, *p* = 0.018.

Figure 3 shows the associations between the Jobe test of aquatic athletes and VAS at rest and during training. A positive result in the Jobe test was significantly associated with a higher score in VAS at rest in aquatic athletes, both females (2.40 ± 2.72 vs. 0.90 ± 1.52; *p* = 0.034; d = 0.680) and males (1.65 ± 1.77 vs. 0.20 ± 0.41; *p* = 0.004; d = 1.129). A positive result in the Jobe test was significantly associated with a higher VAS during training only in male aquatic athletes (4.29 ± 2.54 vs. 2.30 ± 2.58; *p* = 0.036; d = 0.777).

## 4. Discussion

To the best of our knowledge, this is the first study to describe the tendon thicknesses of the RC in sex-paired aquatic athletes, and its relationship with anthropometric and pain parameters. The main findings of the present study are the creation of a US measurement protocol for the RC tendons in overhead athletes, which will serve as a reference for subsequent clinical evaluations or scientific investigations. Following this protocol, our results show higher values in the SSB, SSP, ISP, and Tm tendon thickness and AHD, but not the LHBB tendon thickness, in swimmers and water polo players compared to controls. These findings may imply an adaptive response of the tendon thicknesses to the swimming-induced mechanical load.

Commencing with an examination of the LHBB tendon, few studies have examined its thickness in swimmers and water polo players. Suzuki et al. [1] reported that healthy swimmers and those with shoulder pain had LHBB tendon thickness values of 2.94 mm and 2.85 mm, respectively, while controls exhibited a thickness of 2.66 mm, showing no significant differences among groups. In baseball players, some studies have found that the LHBB tendon thickness was larger in both healthy and injured athletes compared to controls [22], while others have shown a reduction in thickness after pitching due to the tensile stress on the LHBB tendon during this gesture [26]. However, these studies did not account for sex differences; our study reveals that LHBB tendon thickness is significantly thinner in females compared to males in both water polo players and controls. Furthermore, we found that the thickness of all RC tendons is greater in aquatic athletes than in controls, except for the LHBB. This finding is surprising given the substantial biomechanical contribution of this muscle during the push phase of swimming strokes. The absence of adaptive changes may be particularly relevant due to the increased risk of biceps tendinosis in young athletes, which decreases with years of competition [27].

Regarding the SSB tendon, recent studies have identified structural among water polo players compared to swimmers and other throwing athletes [28]. Although in other studies swimmers exhibit greater SSB tendon thickness than controls, no significant differences in thickness have been observed between swimmers with and without pain [1]. Our findings support this, indicating that SSB tendon thickness is greater in aquatic athletes, showing no correlation with pain levels. The increase in the SSB tendon thickness may be an adaptive response to the type of sport, possibly due to the internal rotation that both sports demand, which should be considered when monitoring training load and athletic adaptations.

On the other hand, the SSP tendon is the most extensively investigated in swimming and water polo due to its high incidence of injury. Dischler et al. [29] reported an average thickness of 6.4 mm in collegiate swimmers, with a significant increase in thickness observed among those with several years of competition experience. Both the duration of competition and training affect the tendon thickness, as found by Porter et al. [8], who observed that the SSP tendon thickness increased after high-volume and high-intensity training sessions, particularly in swimmers with shoulder pain, with the effect persisting for up to six hours [30]. In contrast, following a fatigue protocol, the thickness can remain elevated for up to 24 h [16]. However, our results indicate that increased tendon thickness is not directly linked to shoulder pain during rest or training. This thickening may result from the elevated production of proteoglycans and glycosaminoglycans in the tendon matrix, causing more water retention [31]. This phenomenon may result in the tendon occupying a larger volume in the subacromial space, thereby elevating the risk of compression, degeneration and rupture [8]. Furthermore, the accumulation of these substances has also been observed with tendon aging and degeneration [8,27]. An alternative hypothesis suggests that the continuous mechanical loading of the tendon may induce alterations in the extracellular matrix, thereby influencing the diameter and organization of collagen fibers. Such changes could diminish the tendon’s capacity to withstand tensile forces, subsequently promoting degeneration and pain [31]. Studies have also linked higher competition levels to an increased risk of SSP tendinopathy in swimmers, as well as greater tendon thickness in individuals with SIS [11,17]. Dubé et al. [32] found that the SSP tendon thickness normalizes with the contralateral side, regardless of the rehabilitation modality. Therefore, monitoring excessive thickening of the SSP tendon between training sessions is advisable as a strategy to mitigate the risk of shoulder injuries. However, there is a lack of studies that examine the thickness of all RC tendons and their fascia in sports such as swimming or water polo. Therefore, SSP tendon measurements should be considered in conjunction with the other RC tendons. A comprehensive analysis of shoulder anatomy can help diagnose SSP injury and guide the rehabilitation process.

Our findings indicate that SSP tendon thickness is significantly greater in swimmers and water polo players compared to controls, with no significant differences observed between sexes. Although some studies suggest that swimmers with shoulder pain exhibit greater SSP tendon thickness than those without [1], other research has not established a consistent relationship between injury, pain, and tendon thickness [9,14,28,33]. The lack of consensus on the link between SSP tendon thickness and shoulder pain makes it difficult to draw firm recommendations. However, the evident increase in RC tendons with years of swimming training denotes an adaptive response to overhead aquatic sports, which should be considered in further research.

Regarding other tendons, notably the ISP and Tm, the existing literature on their thickness is scarce. Our study revealed greater ISP and Tm tendon thicknesses in swimmers and water polo athletes compared to controls, with females exhibiting thinner ISP tendons than males in both sports. Further investigation into these shoulder tendons is needed, as their adaptation to swim training may be linked to shoulder pathologies. At a muscular level, the activation of the ISP muscle while swimming is greater when there is shoulder pain [34]. Therefore, the execution of scapulohumeral exercises involving the alignment of the head to reduce the imbalance between the RC muscles may be highly recommended [35].

These stabilizing muscles of the shoulder are also essential to maintain adequate AHD during shoulder elevation, preventing humeral head displacement [36]. Training fatigue can disrupt this stabilization, leading to potential impingement [27]. Our findings indicate greater AHD in aquatic athletes compared to controls, with no differences observed between dominant and non-dominant sides. Existing research indicates that the normal range for AHD in a neutral position is between 10–15 mm, with values below 7 mm indicative of potential pathological conditions [37,38]. Our results showed all groups within this range, with male aquatic athletes exceeding control values. Likewise, Özçadırcı et al. [36] reported that lower AHD and increased SSP tendon thickness were associated with shoulder pain in preadolescent swimmers. According to our results, female water polo players exhibited lower AHD than males, particularly during abduction and post-weight training, indicating fatigue-related effects [16]. Additionally, female overhead athletes had a higher AHD on their dominant sides compared to their non-dominant sides, aligning with findings from other studies conducted on overhead female athletes [39]. Elite athletes generally maintain a higher AHD during abduction than recreational athletes [39], while non-elite athletes may experience muscle imbalances that increase the risk of shoulder pain [16,36]. For this reason, it seems important to maintain an AHD within these ranges by strengthening the depressor muscles of the humerus to prevent the rubbing and pinching of the structures that pass through this space.

Despite the biomechanical variations between each arm in the sports analyzed, the swimmers exhibited no significant differences in the thickness of the RC tendons on either the breathing or non-breathing sides. In agreement, Suzuki et al. [1] noted similar LHBB, SSP and SSB tendon thicknesses between dominant and non-dominant swimmers. On the contrary, in water polo, comparative studies using MRI have shown that water polo players are more susceptible to injuries in the SSB and ISP tendons, as well as the posterior labrum of the throwing arm, compared to swimmers and other throwing sports [28,40]. Although no significant differences in SSP tendon thickness have been reported between the throwing and non-throwing sides [28], some studies in overhead baseball athletes found such differences [41]. Our results indicated that the non-throwing side had greater Tm tendon thickness in female water polo players and LHBB tendon thickness in male water polo players. This suggests that the non-throwing arm plays a stabilizing role in the water, while the throwing arm may experience increased friction due to high abduction and flexion during throws. Additionally, anterior instability may arise from the anteroinferior translation of the humerus during the throwing motion [42]. This biomechanical characteristic must be considered by coaches when performing dryland training, to maintain the alignment of both upper extremities.

In terms of shoulder pain, we evaluated it using three commonly used scales and compared the pain levels with findings from shoulder US scans. In particular, females had a lower prevalence of positive results on the Jobe test, which is the most provocative test for shoulder injury, compared to males. These results are consistent with the study of Tate et al. [43], in which 27–52% of swimmers were positive. The present study did not find a correlation between tendon thicknesses in RC and shoulder pain. Therefore, shoulder rehabilitation should prioritize a thorough physical evaluation rather than focusing solely on structural pathology, as previously recommended [35]. Despite this, rehabilitation programs aimed at restoring tendon homeostasis could still benefit from measurements of tendon thickness using the US as an objective method. To ensure effective rehabilitation, a comprehensive analysis of the entire shoulder structure by the clinicians and physiotherapist—including measurements of the LHBB and RC tendons—can provide valuable information on shoulder conditions and enhance the approach to exercise rehabilitation.

Further research is required, stratified by population demographics, gender and age cohorts, to enable its application in clinical examinations of shoulder assessments. Finally, our data have some limitations such as the restricted sample size and the absence of longitudinal data. Future investigations should consider longitudinal monitoring of the shoulder rotator cuff tendons during competitive periods.

## 5. Conclusions

The rotator cuff tendon structures were larger in the overhead aquatic athletes, both swimmers and water polo players, compared to the age- and sex-paired controls, except for the long-head biceps brachii tendon. These results indicate an adaptation of the shoulder structures to athletic movements, which can be useful in monitoring athletes’ training load throughout the tendon thickness enlargement.

The females had thinner shoulder tendons than the age- and sport-paired males, except for the supraspinatus tendon, and a lower incidence of positive Jobe test results compared to the males, which is associated with lower pain at rest and during training.

This article provides reference values for ultrasound measurements of shoulder tendons for future clinical examinations of swimmers and water polo athletes, as well as a comprehensive measurement protocol that may serve as a foundation for subsequent studies, thereby facilitating a more precise comparison of tendon thicknesses.

## Figures and Tables

**Figure 1 healthcare-13-00151-f001:**
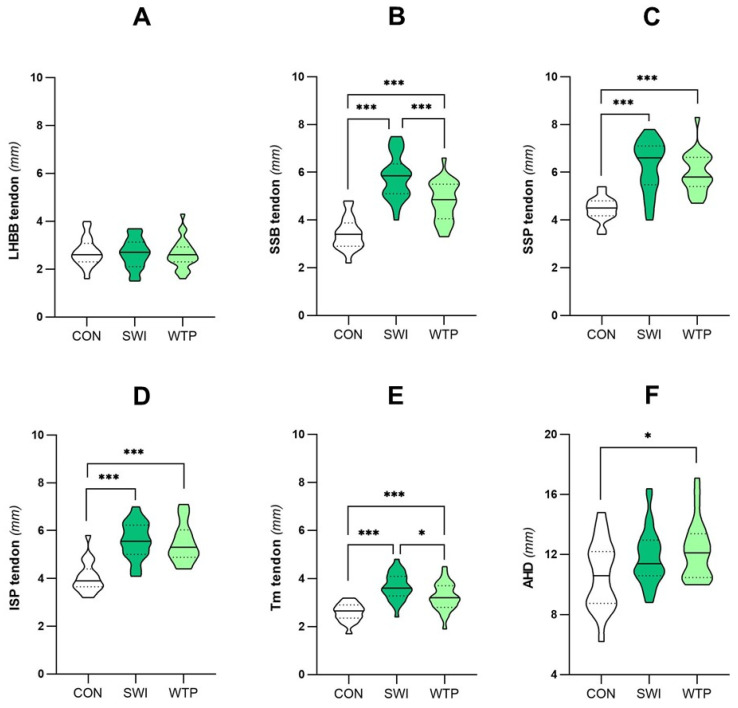
Comparison of the shoulder ultrasonographic parameters between controls (CON), swimmers (SWI) and water polo players (WTP) in female participants. The differences in the mean thickness of the rotator cuff tendons studied and the acromiohumeral distance in females are represented, according to the group to which they belong. The analysis was on the following structures: (**A**) Long-head biceps brachii (LHBB) tendon; (**B**) subscapularis (SSB) tendon; (**C**) supraspinatus (SSP) tendon; (**D**) infraspinatus (ISP) tendon; (**E**) teres minor (Tm); and (**F**) acromiohumeral distance (AHD). Statistically significant differences are denoted by * in case of *p* < 0.05 and *** in case of *p* < 0.001.

**Figure 2 healthcare-13-00151-f002:**
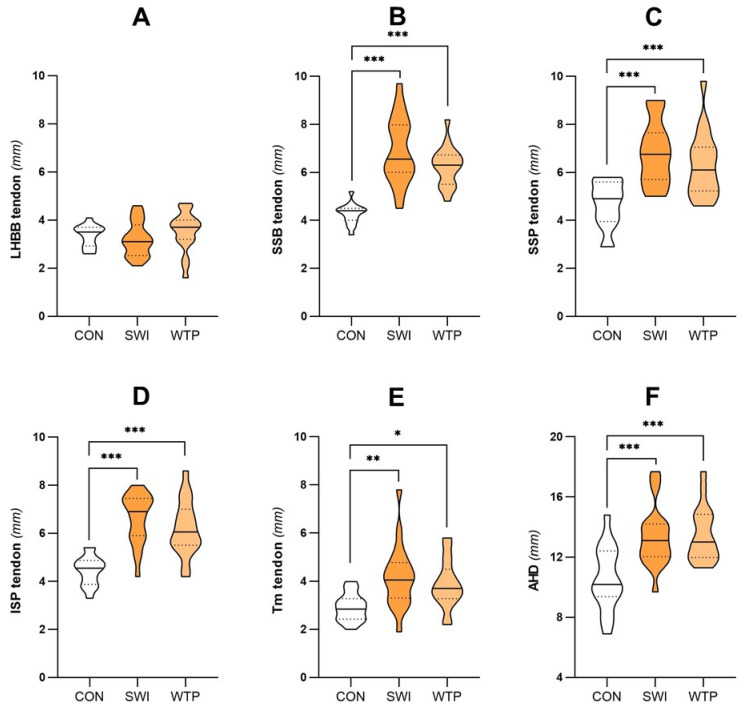
Comparison of the shoulder ultrasonographic parameters between controls (CON), swimmers (SWI) and water polo players (WTP) in male participants. The differences in the mean thickness of the rotator cuff tendons studied and the acromiohumeral distance in males are represented, according to the group to which they belong. The analysis includes: (**A**) The long-head biceps brachii (LHBB) tendon; (**B**) subscapularis (SSB) tendon; (**C**) supraspinatus (SSP) tendon; (**D**) infraspinatus (ISP) tendon; (**E**) teres minor (Tm); and (**F**) acromiohumeral distance (AHD). Statistically significant differences are denoted by * in case of *p* < 0.05, ** in case of *p* < 0.01, and *** in case of *p* < 0.001.

**Figure 3 healthcare-13-00151-f003:**
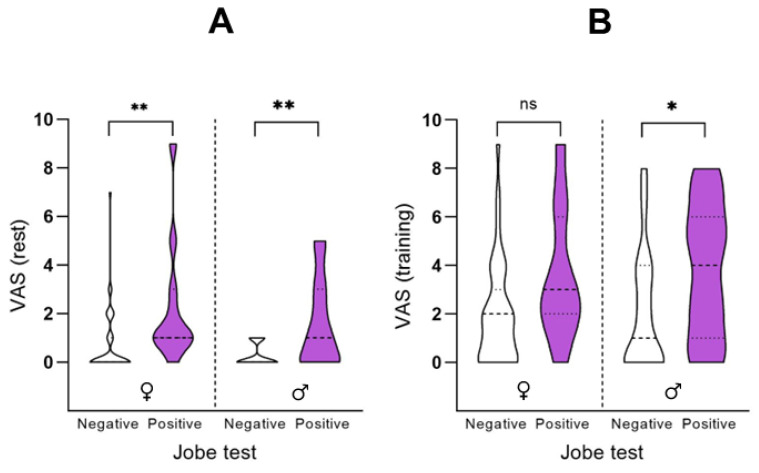
Relationship between Jobe test result (positive or negative) and pain scales in female and male aquatic athletes. The Figure represents the relationship between a positive or negative result in the Jobe test and the mean score obtained on the visual analogue scale at rest and during training, divided by sex. (**A**) Visual Analogue Scale (VAS) at rest; and (**B**) VAS during training. Statistically significant differences are denoted by * in case of *p* < 0.05, and ** in case of *p* < 0.01.

**Table 1 healthcare-13-00151-t001:** Anatomic references of the tendons and fascia muscles from the shoulder measured by ultrasound and anatomic techniques.

Muscle	Anatomic Reference	US Image	US Transducer
SSB tendon	Long axis: The midpoint between the LT of the humerus and the beginning of JC. Arm at 45° of external rotation.	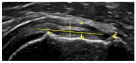	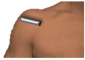
LHBB tendon	Short axis: US image at the highest and most defined point of the LT and GT. Arm with supinated hand resting on thigh.	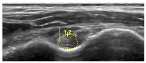	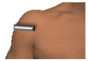
SSP tendon	Short axis: The averageof the measures taken at 10, 15 and 20 mm lateral to the biceps tendon. Hand on the ipsilateral posterior hip [17].	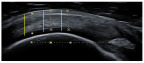	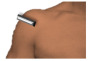
ISP tendon	Long axis: Transducer below the level of the posterior surface of the acromion, parallel to the ISP. Hand of the recruit placed on the contralateral shoulder [22].	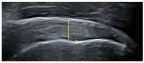	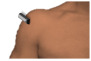
Tm tendon	Short axis: The midpoint between the LT’s inferior facet and the JC’s beginning. Arm in the contralateral shoulder.	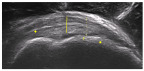	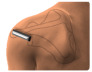
SSP fascia	Short axis: The perpendicular point to the deepest area of the SSPF. Arm resting in a neutral position.	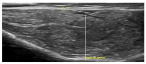	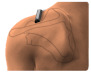
ISP fascia	Short axis: The perpendicular point to the deepest area of the ISPF. Arm resting in a neutral position.	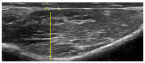	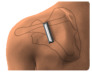
Tm fascia	Short axis: The perpendicular point to the deepest area of the TmF. Arm resting in a neutral position.	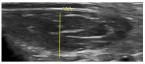	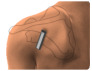

The measurement points for data collection using the ultrasound technique are described. US: ultrasound, SSB: subscapularis, LHBB: long head of bices brachii, SSP: supraspinatus, ISP: infraspinatus, Tm: teres minor, LT: lesser tubercle, GT: greater tubercle JC: joint capsule, SSPF: supraspinatus fossa, ISPF: infraspinatus fossa, TmF: teres minor fossa. Modified image from Arrillaga et al. (2024) [23].

**Table 2 healthcare-13-00151-t002:** Sex differences in the ultrasonographic shoulder parameters of controls, swimmers and water polo players.

	Controls		Swimmers		Water Polo Players	
	Females (*n* = 18)	Males (*n* = 16)	Females (*n* = 30)	Males (*n* = 20)	Females (*n* = 26)	Males (*n* = 18)
LHBB tendon *(mm)*	2.72 ± 0.60	3.35 ± 0.45 **	2.66 ± 0.61	3.21 ± 0.71	2.63 ± 0.61	3.63 ± 0.69 **
SSB tendon *(mm)*	3.43 ± 0.68	4.27 ± 0.43 **	5.81 ± 0.91	6.91 ± 1.36	4.79 ± 0.83	6.27 ± 0.87 **
SSP tendon *(mm)*	4.47 ± 0.50	4.67 ± 0.95	6.24 ± 1.06	6.81 ± 1.30	5.99 ± 0.84	6.31 ± 1.35
ISP tendon *(mm)*	4.05 ± 0.66	4.46 ± 0.60	5.58 ± 0.77	6.63 ± 1.04 **	5.48 ± 0.81	6.21 ± 1.14 *
Tm tendon *(mm)*	2.58 ± 0.39	2.92 ± 0.57 *	3.67 ± 0.56	4.19 ± 1.29	3.25 ± 0.58	3.87 ± 0.99
SSP fascia *(mm)*	0.62 ± 0.11	0.64 ± 0.12	0.61 ± 0.19	0.59 ± 0.15	0.61 ± 0.14	0.62 ± 0.13
ISP fascia *(mm)*	0.62 ± 0.14	0.61 ± 0.12	0.47 ± 0.15	0.50 ± 0.14	0.51 ± 0.11	0.46 ± 0.12
Tm fascia *(mm)*	0.55 ± 0.11	0.61 ± 0.14	0.50 ± 0.14	0.48 ± 0.12	0.53 ± 0.10	0.51 ± 0.15
AHD *(mm)*	10.62 ± 2.31	10.58 ± 2.11	11.83 ± 1.83	13.39 ± 2.01	12.30 ± 1.92	13.48 ± 1.80 **
OR *(SSP/AHD)*	0.44 ± 0.10	0.45 ± 0.09	0.52 ± 0.12	0.48 ± 0.12	0.43 ± 0.21	0.47 ± 0.09

The table expresses the average thickness of the tendons of the muscles and fascias of the shoulder studied, as well as the acromiohumeral distance and the occupation ratio, in males and females, and according to the group to which they belong. The values are expressed by as mean ± standard deviation (SD). Significant differences by sexes within-groups are denote by * (*p* < 0.05), and ** (*p* < 0.01). LHBB: long head of biceps brachii, SSB: subscapularis, SSP: supraspinatus, ISP: infraspinatus, Tm: teres minor, AHD: acromiohumeral distance, OR: occupation ratio.

**Table 3 healthcare-13-00151-t003:** Shoulder ultrasonographic parameters in swimmers, comparing breath and non-breathing side.

	Swimmers (*n* = 36)			
	Females (*n* = 20)		Males (*n* = 16)	
	Breathing side	Non-breathing side	Breathing side	Non-breathing side
LHBB tendon *(mm)*	2.94 ± 0.54	2.89 ± 0.60	3.56 ± 0.72	3.06 ± 0.46
SSB tendon *(mm)*	5.41 ± 0.92	6.07 ± 1.07	6.35 ± 1.01	6.8 ± 1.4
SSP tendon *(mm)*	6.06 ± 1.20	6.31 ± 1.19	6.43 ± 1.03	6.68 ± 1.57
ISP tendon *(mm)*	5.09 ± 0.69	5.48 ± 0.71	6.08 ± 1.15	6.63 ± 0.72
Tm tendon *(mm)*	3.67 ± 0.64	3.29 ± 0.23	4.30 ± 1.15	4.09 ± 1.78
SSP fascia *(mm)*	0.67 ± 0.18	0.55 ± 0.20	0.61 ± 0.17	0.58 ± 0.13
ISP fascia *(mm)*	0.47 ± 0.15	0.47 ± 0.16	0.51 ± 0.15	0.49 ± 0.14
Tm fascia (mm)	0.49 ± 0.14	0.50 ± 0.13	0.46 ± 0.09	0.50 ± 0.15
AHD *(mm)*	12.33 ± 1.94	12.02 ± 1.96	14.00 ± 2.04	13.70 ± 1.78
OR *(SSP/AHD)*	0.50 ± 0.13	0.53 ± 0.11	0.47 ± 0.10	0.49 ± 0.13

The table expresses the thickness of the tendons of the muscles and fascias of the shoulder, as well as the acromiohumeral distance and the occupation ratio, in female and male swimmers, stratified by the breathing and the non-breathing side. The values are expressed by as mean ± standard deviation (SD). LHBB: long head of biceps brachii, SSB: subscapularis, SSP: supraspinatus, ISP: infraspinatus, Tm: teres minor, AHD: acromiohumeral distance, OR: occupation ratio.

**Table 4 healthcare-13-00151-t004:** Shoulders ultrasonographic parameters in water polo players, comparing throwing arm and non-throwing arm.

	Water Polo Players (*n* = 44)			
	Females (*n* = 26)		Males (*n* = 18)	
	Throwing	Non-throwing	Throwing	Non-throwing
LHBB tendon *(mm)*	2.63 ± 0.53	2.61 ± 0.64	3.24 ± 0.66	4.02 ± 0.50 *
SSB tendon *(mm)*	4.95 ± 0.65	4.84 ± 0.90	0.62 ± 0.07	0.64 ± 0.09
SSP tendon *(mm)*	6.16 ± 0.95	5.81 ± 0.72	6.66 ± 1.39	5.96 ± 1.28
ISP tendon *(mm)*	5.69 ± 0.93	5.27 ± 0.65	6.23 ± 0.87	6.18 ± 1.41
Tm tendon *(mm)*	3.10 ± 0.31	3.53 ± 0.56 *	3.97 ± 1.24	3.77 ± 0.72
SSP fascia *(mm)*	0.65 ± 0.17	0.60 ± 0.12	0.63 ± 0.15	0.61 ± 0.11
ISP fascia *(mm)*	0.53 ± 0.13	0.49 ± 0.10	0.47 ± 0.16	0.44 ± 0.09
Tm fascia *(mm)*	0.52 ± 0.12	0.54 ± 0.10	0.54 ± 0.17	0.48 ± 0.13
AHD *(mm)*	11.70 ± 1.55	12.66 ± 2.23	12.72 ± 1.23	14.24 ± 2.01
OR *(SSP/AHD)*	0.53 ± 0.10	0.41 ± 0.20	0.52 ± 0.09	0.42 ± 0.06

The table expresses the thickness of the tendons of the muscles and fascias of the shoulder, as well as the acromiohumeral distance and the occupation ratio, in female and male water polo players, stratified by the throwing side or the non-throwing side. The values are expressed as mean ± standard deviation (SD). Significant differences by sexes are denoted by * (*p* < 0.05). LHBB: long-head of biceps brachii, SSB: subscapularis, SSP: supraspinatus, ISP: infraspinatus, Tm: teres minor, AHD: acromiohumeral distance, OR: occupation ratio.

## Data Availability

The data are available upon reasonable request to D.L.S. (mimiguel@ub.edu).

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
