# Peer review of "Swimmer’s Shoulder: Ultrasound Anatomical Description of Shoulder Tendons in Elite Swimmers and Water Polo Players"

_healthcare, 2025, doi:10.3390/healthcare13020151_

Round 1
Reviewer 1 Report
Comments and Suggestions for Authors
This study represents a valuable contribution to the understanding of shoulder biomechanics in swimmers and water polo players. The rigorous methodology, clear writing style, and clinically relevant findings make this a strong study. The inclusion of sex-specific analysis adds to its impact. The overall findings highlight the need for further investigation into the relationship between tendon structure and function, including pain, in these athletes. Some minor reviews should be addresed:
Methods: Participants: The sample size (64 junior athletes, 128 shoulders) is clearly stated, along with the breakdown into groups (controls, swimmers, water polo players). The description of the athletes as "elite junior level" needs further clarification regarding selection criteria. The inclusion of FINA points for swimmers is useful for defining athletic level. The description of the training regimens for each group provides valuable context. However, more detail is needed on how participants were recruited. Was there a specific inclusion or exclusion criteria? Was there any attempt to match participants in the different groups beyond age and sex? The mention of ethical approval and informed consent is important and adheres to ethical standards.
Data Analysis: The statistical methods are adequately described, including the use of Shapiro-Wilk test, two-way repeated-measures ANOVA, Bonferroni correction, effect size calculations (η²p and Cohen's d), and chi-square test. The authors need to clarified which specific variables are included in the analysis. The mention of statistical power is also a positive aspect. Is there a power analysis included in the study?
Results: The results section is adequately structured, and it was detailed and comprehensive. The authors should insert further details of statistical analyses and the numerical results should be reported within the text, allowing for the reader to understand the findings without resorting to the tables.
Author Response
Dear Editor and Reviewers,
Thank you very much for the valuable suggestions and constructive assessments on our manuscript “The swimmer's shoulder: Ultrasound anatomical description of shoulder tendons in elite swimmers and water polo players”. All your comments have been taken into consideration, and the changes or amendments introduced in the text are detailed in the point-to-point answers below in addition to the revised manuscript.
Comment: “Methods: Participants: The sample size (64 junior athletes, 128 shoulders) is clearly stated, along with the breakdown into groups (controls, swimmers, water polo players). The description of the athletes as "elite junior level" needs further clarification regarding selection criteria. The inclusion of FINA points for swimmers is useful for defining athletic level. The description of the training regimens for each group provides valuable context. However, more detail is needed on how participants were recruited. Was there a specific inclusion or exclusion criterion?
Answer: Thank you very much for your concise review of the methodology. Athletic Classification Frameworks such as McKay et al. (2022) are based on adult athletes, without considering age categories. Our sample (~15 years old) can be classified in Tier 4 “Elite/International level” since the participants are likely to be members of the national junior swimming team, some of them are international medallists and they are part of a training program with intention to complete at top-level competition. The inclusion and exclusion criteria considered when recruiting participants have been included in lines 118-129.
Comment: Was there any attempt to match participants in the different groups beyond age and sex? The mention of ethical approval and informed consent is important and adheres to ethical standards”.
Answer: Thank you for your question. The participants were all part of the same training developmental program. Therefore, they were healthy swimmers as it is stated now in the Inclusion Criteria, they have the same age and were part of the same training program. Therefore, only the difference between sex and sport was considered.
Comment: “Data Analysis: The statistical methods are adequately described, including the use of Shapiro-Wilk test, two-way repeated measures ANOVA, Bonferroni correction, effect size calculations (η²p and Cohen's d), and chi-square test. The authors need to clarify which specific variables are included in the analysis. The mention of statistical power is also a positive aspect. Is there a power analysis included in the study?”.
Answer: Thank you for the valuable feedback. We have included some more explanations regarding the statistical analysis in lines 113 to 117.
Participants were divided by sex due to (i) the already-known anatomical differences between females and males, and (ii) to enhance the clearness of the results. Then, we conducted a two-way repeated measures ANOVA, in which sex (females and males) and group (controls, swimmers and water polo players were considered as the independent factors, as said in line 154.
The dependent variables of the study were the thickness of the tendons studied: tendon of the subscapularis muscles, the long head of the biceps brachii, supraspinatus, infraspinatus and teres minor; and the thickness of the fascias studied: the thickness of the supraspinatus, infraspinatus and teres minor muscles fascias. This has been clarified in line 153.
Regarding the power analysis, we estimated the minimum sample size considering the lowest effect size resulting in statistically significant results of a shoulder parameter (0.292). Then, we computed the required sample size for a two-way repeated measures ANOVA (2x3), in which an effect size of 0.292, an alpha level of 0.05, as well as a power of 0.90, resulting in a total sample size of 42, which was smaller than the 128 shoulders evaluated.
Comment: “Results: The results section is adequately structured, and it was detailed and comprehensive. The authors should insert further details of statistical analyses, and the numerical results should be reported within the text, allowing for the reader to understand the findings without resorting to the tables.”
Answer: Thank you. The numerical results have been described now in the corresponding section, in addition to the Table/Figure.

Reviewer 2 Report
Comments and Suggestions for Authors
Dear Editor,
The manuscript is well written, but some improvements need to be made to strengthen the research article.
Thank you

Author Response
Dear Editor and Reviewers,
Thank you very much for the valuable suggestions and constructive assessments on our manuscript “The swimmer's shoulder: Ultrasound anatomical description of shoulder tendons in elite swimmers and water polo players”. All your comments have been taken into consideration, and the changes or amendments introduced in the text are detailed in the point-to-point answers below in addition to the revised manuscript.
Comment 1: Introduction: “The manuscript is well written, but some improvements need to be made. The problem in the research is already there but we suggest strengthening it based on data”.
Answer: Thank you for the valuable review. A more comprehensive description of the background and justification of the study has been included in the Introduction, considering biomechanical and fatigue aspects which are decisive in the study of the thickness of the rotated cuff tendons, as well as shoulder pain.
Comment 2: Conclusions: “Add research recommendations”.
Answer: We appreciate your comment. New lines of research recommendations and considerations have been included in this section.

Reviewer 3 Report
Comments and Suggestions for Authors
The paper examines the differences in shoulder ultrasonographic parameters among swimmers, water polo players, and controls and their relationship with shoulder pain in sex- and age-paired groups. The study also seeks to provide reference values to inform the prevention and rehabilitation of shoulder pain in overhead aquatic athletes.
Introduction
The introduction could better integrate the relationship between tendon adaptations and long-term injury prevention in aquatic sports. It mainly focuses on tendon thickness without linking it to biomechanical or physiological consequences. Please provide a more detailed rationale for measuring tendon thickness, which is critical for managing athlete health. Also, paragraphs are too short; a paragraph must be longer than two sentences.
Materials and Methods
The study does not fully account for potential confounding factors such as variations in training intensity and techniques among athletes and differences in injury history or chronic shoulder conditions. Please include a more detailed analysis or discussion of how these variables influenced the results. Descriptive parameters of the sample should be included in this section.
L119:121 should be corrected to: "Effect sizes as partial eta squared (ηp2\eta^2_p) values were employed to present the magnitude of differences, with thresholds of 0.01, 0.06, and 0.14 for small, medium, and large effects, respectively."
*Pearson's linear regression (measures the relationship between continuous variables) and chi-square tests (assesses the association between categorical variables)* Please correct.
Discussion
The discussion highlights adaptive responses but lacks integration with practical implications for training or rehabilitation. There is insufficient exploration of the lack of correlation between tendon thickness and shoulder pain, which contradicts some existing literature. Please expand the discussion to include practical applications of findings for coaches and clinicians and potential reasons for the lack of correlation between thickness and pain, referencing existing biomechanical or physiological theories.
The study includes elite junior athletes. The findings may not be generalizable to recreational or older athletes. Include a discussion on the limitations of generalizability and consider expanding future research to more diverse groups.
Figures and Tables
The figures (e.g., ultrasonographic measurements) lack clear legends and explanations, which might hinder understanding for non-specialists. Tables summarizing sex and group comparisons do not highlight key findings sufficiently. Please enhance figure legends and provide clearer, more focused tables summarizing statistically significant findings.
Comments on the Quality of English Language
It should be improved.
Author Response
Dear Editor and Reviewers,
Thank you very much for the valuable suggestions and constructive assessments on our manuscript “The swimmer's shoulder: Ultrasound anatomical description of shoulder tendons in elite swimmers and water polo players”. All your comments have been taken into consideration, and the changes or amendments introduced in the text are detailed in the point-to-point answers below in addition to the revised manuscript. The English language has been reviewed to enhance the clarity.
Introduction:
Comment 1: “The introduction could better integrate the relationship between tendon adaptations and long-term injury prevention in aquatic sports. It mainly focuses on tendon thickness without linking it to biomechanical or physiological consequences. Please provide a more detailed rationale for measuring tendon thickness, which is critical for managing athlete health. Also, paragraphs are too short; a paragraph must be longer than two sentences”.
Answer: Thank you very much for your valuable review. The introduction has been improved to include more comprehensive details of the mechanical aspects of the sporting gesture, as well as the reason why monitoring tendon thickness may be relevant to preventing long-term injuries.
Currently, there are no studies that examine, beyond 24 hours post-training, the changes in the tendon of the rotator cuff tendons over time, and whether they have been related to the increase or not in shoulder injuries. For this reason, baseline measurements may be a starting point to conduct further examinations and its relationship with injuries.
Materials and Methods:
Comment 2: “The study does not fully account for potential confounding factors such as variations in training intensity and techniques among athletes and differences in injury history or chronic shoulder conditions. Please include a more detailed analysis or discussion of how these variables influenced the results. Descriptive parameters of the sample should be included in this section”.
Answer: We appreciate your feedback. Now, we have included the Inclusion and Exclusion criteria that were considered for the participation of the subjects in the Material and Methods section. They were healthy athletes with no previous shoulder surgery in their lifetime, or any shoulder injury in the last 6 months.
The potential relationship between shoulder pain and tendon thicknesses was considered. The results in the VAS scale and its relationship with tendon thickness did not show a significant relationship, as it says in lines 331-336.
Regarding the variations in the training intensity and techniques, all subjects belong to the same training developmental group, so they perform the same hours, kilometres and swimming modalities during the week, both swimmers and water polo players. In the case of swimmers, although each one is getting more specialized in a swimming modality when competing, they are still in a developmental program so the training routine remains very similar between them. In the case of water polo players, the participants are part of the same team, so they have a very similar training routine, but we agree that there may be some differences depending on the position they occupy on the team (e.g. goalkeeper). It would be interesting to compare whether there are differences along the positions they play with a larger sample size. This comment has been added to the Discussion section.
Comment 3: “L119:121 should be corrected to: "Effect sizes as partial eta squared (ηp2\eta^2_pηp2​) values were employed to present the magnitude of differences, with thresholds of 0.01, 0.06, and 0.14 for small, medium, and large effects, respectively".”.
Answer: Thank you. Corrected.
Comment 4: “*Pearson's linear regression (measures the relationship between continuous variables) and chi-square tests (assesses the association between categorical variables)* Please correct”.
Answer: Ok. We have included this information in the Statistical Analysis of the manuscript.
Discussion:
Comment 5: “The discussion highlights adaptive responses but lacks integration with practical implications for training or rehabilitation. There is insufficient exploration of the lack of correlation between tendon thickness and shoulder pain, which contradicts some existing literature. Please expand the discussion to include practical applications of findings for coaches and clinicians and potential reasons for the lack of correlation between thickness and pain, referencing existing biomechanical or physiological theories”.
Answer: Thank you very much for the valuable comment. We have included a more comprehensive analysis of the existing literature in the Discussion section, including some relevant insights in terms of the relationship between shoulder pain and tendon thickness, as well as practical applications for coaches and shoulder pain.
Comment 6: “The study includes elite junior athletes. The findings may not be generalizable to recreational or older athletes. Include a discussion on the limitations of generalizability and consider expanding future research to more diverse groups”.
Answer: We appreciate your comment, and we agree with it. The results can be a landmark when comparing with subjects of the same physical fitness, age and groups. Based on another study carried out in my research group on cadaveric specimens, the thickness measurements cannot be compared with such different groups and even, as the results of this study show, there is already a difference between athletes and control subjects of the same age. The possibility of continuing along this future line of research will be added to the article in the conclusions.
Figures and Tables:
Comment 7: “The figures (e.g., ultrasonographic measurements) lack clear legends and explanations, which might hinder understanding for non-specialists. Tables summarizing sex and group comparisons do not highlight key findings sufficiently. Please enhance figure legends and provide clearer, more focused tables summarizing statistically significant findings”.
Answer: Thank you for the suggestion. A clearer description of what each graph represents has been included in the tables and figures, for a better understanding.

Round 2
Reviewer 3 Report
Comments and Suggestions for Authors
Introduction
Contextualization: The introduction sets the context but could briefly highlight the clinical significance of understanding tendon adaptations in overhead aquatic athletes.
Research Gap: The gap is identified, but a clearer justification for the study's necessity (e.g., "lack of data on sex-based differences in tendon thickness") can make the argument more compelling.
Objectives: Consider rephrasing the objectives to be more specific and measurable.
Materials and Methods
Participant Description: The criteria for inclusion and exclusion are clear, but a brief explanation of how "healthy controls" were screened for shoulder pathologies would be helpful.
Discussion
Critical Analysis: The discussion could delve deeper into the implications of findings, especially regarding training practices or injury prevention strategies.
Integration with Literature: While the discussion references relevant studies, it could benefit from comparing results with non-aquatic athletes or athletes in other overhead sports.
Clinical Recommendations: Explicitly link findings to actionable recommendations for coaches, physiotherapists, or clinicians managing shoulder health in aquatic athletes.
Conclusion
Clarity: Rephrase the conclusion to summarize the findings succinctly, highlight the practical applications, and stress the need for future studies.
Limitations and Future Research: Explicitly outline study limitations (e.g., small sample size, absence of longitudinal data) and propose areas for further investigation.
Writing Style and Structure
Language: Avoid overly technical jargon where unnecessary, and ensure consistent use of abbreviations (e.g., LHBB, SSB) throughout the text.
Flow: Some sections, such as the discussion, could benefit from better paragraph transitions to enhance readability.
Comments on the Quality of English LanguageGood.
Author Response
20-12-2024
Dear Editor and Reviewers,
Thank you very much for the valuable suggestions and constructive assessments on our manuscript “The swimmer's shoulder: Ultrasound anatomical description of shoulder tendons in elite swimmers and water polo players”. All your comments have been taken into consideration, and the changes or amendments introduced in the text are detailed in the point-to-point answers below in addition to the revised manuscript.
Introduction
Contextualization: The introduction sets the context but could briefly highlight the clinical significance of understanding tendon adaptations in overhead aquatic athletes.
Answer: We appreciate your comment. Modifications have been implemented to ensure that the scientific background underscores the clinical relevance of comprehending the thickness measurements of the various tendons within the rotator cuff.
Research Gap: The gap is identified, but a clearer justification for the study's necessity (e.g., "lack of data on sex-based differences in tendon thickness") can make the argument more compelling.
Answer: Ok. The aforementioned information has been incorporated into the Introduction, accompanied by a reorganization of the section.
Objectives: Consider rephrasing the objectives to be more specific and measurable.
Answer: Thank you. The objective of the study has been revised to improve its clarity.
Materials and Methods
Participant Description: The criteria for inclusion and exclusion are clear, but a brief explanation of how "healthy controls" were screened for shoulder pathologies would be helpful.
Answer: Thank you for the comment. The controls were not screened specifically for shoulder pathologies. Their shoulder’s health status was self-reported. As it is stated in the Exclusion criteria, any participant with previous shoulder surgery in their lifetime or any previous shoulder injury in the last 6 months were not considered eligible for the study.
Discussion
Critical Analysis: The discussion could delve deeper into the implications of findings, especially regarding training practices or injury prevention strategies.
Answer: We express our gratitude for your recommendation. We have intensified our efforts to elucidate the significance of our findings from a perspective that can enhance decision-making in the domains of training and injury prevention strategies.
Integration with Literature: While the discussion references relevant studies, it could benefit from comparing results with non-aquatic athletes or athletes in other overhead sports.
Answer: Thank you. Your suggestion has been considered and we have added some comparisons with other overhead sports in lines 377-388, 455-456 and 470.
Clinical Recommendations: Explicitly link findings to actionable recommendations for coaches, physiotherapists, or clinicians managing shoulder health in aquatic athletes.
Answer: Thank you very much for your comment. We have added the corresponding modifications throughout the article, especially in lines 476-478 and 489.
Conclusion
Clarity: Rephrase the conclusion to summarize the findings succinctly, highlight the practical applications, and stress the need for future studies.
Answer: Thank you very much for your comment. The conclusions have been revised to attain greater specificity and to underscore the principal findings of our study in a practical manner.
Limitations and Future Research: Explicitly outline study limitations (e.g., small sample size, absence of longitudinal data) and propose areas for further investigation.
Answer: Thank you. We have included some of these ideas as Limitations and Future Research at the end of the Discussion section.
Writing Style and Structure
Language: Avoid overly technical jargon where unnecessary, and ensure consistent use of abbreviations (e.g., LHBB, SSB) throughout the text.
Answer: Thank you for your comment. The abbreviations and vocabulary in the text have been revised.
Flow: Some sections, such as the discussion, could benefit from better paragraph transitions to enhance readability.
Answer: We appreciate your feedback and have made changes so that the discussion can be understood in a more connected way.
